# Further Developments towards a Minimal Potent Derivative of Human Relaxin-2

**DOI:** 10.3390/ijms241612670

**Published:** 2023-08-11

**Authors:** Thomas N. G. Handley, Praveen Praveen, Julien Tailhades, Hongkang Wu, Ross A. D. Bathgate, Mohammed Akhter Hossain

**Affiliations:** 1The Florey, Melbourne, VIC 3052, Australia; thomas.handley@florey.edu.au (T.N.G.H.); praveen@student.unimelb.edu.au (P.P.); hongkangw@student.unimelb.edu.au (H.W.); 2Department of Biochemistry and Molecular Biology, Monash University, Melbourne, VIC 3004, Australia; julien.tailhades@monash.edu; 3Department of Biochemistry and Pharmacology, University of Melbourne, Melbourne, VIC 3010, Australia; 4School of Chemistry and Bio21, University of Melbourne, Melbourne, VIC 3010, Australia

**Keywords:** human relaxin-2, RXFP1, 7BP, B7-33, B9-31, fibrosis, peptide synthesis

## Abstract

Human relaxin-2 (H2 relaxin) is a peptide hormone with potent vasodilatory and anti-fibrotic effects, which is of interest for the treatment of heart failure and fibrosis. H2 relaxin binds to the Relaxin Family Peptide Receptor 1 (RXFP1). Native H2 relaxin is a two-chain, three-disulfide-bond-containing peptide, which is unstable in human serum and difficult to synthesize efficiently. In 2016, our group developed B7-33, a single-chain peptide derived from the B-chain of H2 relaxin. B7-33 demonstrated poor affinity and potency in HEK cells overexpressing RXFP1; however, it displayed equivalent potency to H2 relaxin in fibroblasts natively expressing RXFP1, where it also demonstrated the anti-fibrotic effects of the native hormone. B7-33 reversed organ fibrosis in numerous pre-clinical animal studies. Here, we detail our efforts towards a minimal H2 relaxin scaffold and attempts to improve scaffold activity through Aib substitution and hydrocarbon stapling to re-create the peptide helicity present in the native H2 relaxin.

## 1. Introduction

Fibrosis is a pathological feature of most cardiovascular and chronic inflammatory diseases (such as heart failure and chronic obstructive pulmonary disease (COPD)), which are significant causes of global death [1]. Fibrosis is effectively aberrant wound healing, which contributes to organ dysfunction and ultimately failure through the build-up of extracellular wound healing components [2]. Current anti-fibrotic therapies only delay disease progression and can induce severe side effects after chronic administration of drugs [3]. The human gene-2 relaxin peptide (H2 relaxin, Figure 1A) has strong anti-fibrotic [4], vasodilatory [5,6], and cardioprotective effects [7,8]; as a result, H2 relaxin is undergoing/has undergone clinical and pre-clinical development by several pharmaceutical companies (Relaxera, Novartis, and AstraZeneca to name a few) for its anti-fibrotic and cardioprotective capacities mediated by the activation of its cognate G-protein-coupled receptor, RXFP1 [4,9,10]. While it possesses enormous promise as a drug, H2 relaxin has a short in vivo half-life (minutes) and—as a two (A and B)-chain, three-disulfide bond peptide (Figure 1A)—is difficult and expensive to produce and modify [11]. It also cross-reacts with RXFP2, a native receptor for the related peptide INSL3 [12], with currently unknown consequences. H2 relaxin’s strong activation of cAMP signaling may also induce side effects, including tumor growth, in long-term use [13]. Excessive cAMP activation in the ischemic heart could also potentially increase heart rate and myocardial workload [14], further exacerbating the contractile demand on myocytes [15]. In 2016, we reported a potent single-B-chain analog of H2 relaxin, B7-33 (Figure 1C) [13]. B7-33 consists of residues 7–29 of the B chain of H2 relaxin (Figure 1B,C), with the addition of a KRSL sequence from positions 30–33 of the B1-33 isoform of H2 relaxin, which dramatically improves solubility over B7-29 [13]. B7-33, being a single-chain peptide, is far easier and cheaper to produce than the parent peptide, making it an excellent scaffold for modification with the aim of improving its pharmacological properties as an anti-fibrotic therapeutic [13] and a component of anti-fibrotic device coating [16]. While B7-33 has poor affinity and cAMP potency in HEK cells overexpressing RXFP1 (HEK-RXFP1), it retains the known beneficial effects of H2 relaxin (e.g., anti-fibrotic, vasodilatory, anti-inflammatory, and anti-hypertrophy) [13,17,18]. Our published data have demonstrated that B7-33 is equipotent to H2 relaxin in reducing fibrosis in four pre-clinical models of cardiovascular and lung disease [13,17,18]. In a more recent study, we demonstrated that B7-33 can induce more rapid cardioprotective effects over the ACE inhibitor perindopril in mice with dilated cardiomyopathy [19]. Importantly, unlike H2 relaxin, B7-33 did not exacerbate prostate tumor growth in mice, which is postulated to be a cAMP-mediated side-effect [13]. More recently, stabilized variants of B7-33 have been reported by the pharmaceutical company, Sanofi [20,21]. While these analogs are shown to be long-acting in vivo, they regained cAMP potency, similar to H2 relaxin, that could have detrimental effects (such as tumor growth and increased heart rate). This study, therefore, focuses on developing B7-33 analogs that retain the beneficial effects of the B7-33 parent peptide in addition to low potency for cAMP activation. 

## 2. Results and Discussion

### 2.1. Identification of Key Residues for Binding to 7BP

In competitive binding assays in HEK-RXFP1 cells, B7-33 shows a low binding affinity for RXFP1 (Figure 2), ultimately making comparisons of B7-33 analogs difficult. We hypothesized that B7-33 must have a higher affinity for RXFP1 in native cells, but the expression levels in these cells are not compatible with the competition binding assay. We therefore tested the binding affinity of B7-33 in HEK cells expressing an engineered RXFP1 ectodomain construct, 7BP, which contains the RXFP1 ectodomain fused to the single trans-membrane-spanning domain of CD8 (Figure 2B) [22]. We have previously used this construct to unmask the anticipated high-affinity binding of relaxin analogs to the RXFP1 ectodomain [23,24], and here, we used it to determine the pKi of B7-33 (Figure 2C). The binding affinity of B7-33 in HEK-7BP cells is markedly greater than in HEK-RXFP1 cells, reflecting B7-33’s ability to bind strongly to the RXFP1’s ectodomain.

H2 relaxin and RXFP1 have been extensively studied through Structure Activity Relationship (SAR) studies, and the key residues for relaxin binding to the LRRs are R^B13^, R^B17^, and I^B20^ (Figure 2) [25,26]. In our earlier publication on B7-33, we explored the effect of R13A, R17A, and I20A mutations on binding to the native RXFP1 receptor, which showed that loss of any of these key residues abolished the binding to RXFP1 within the limits tested [13]. Here, we analyzed the effect of these same mutations in B7-33 in our HEK-7BP competitive binding assay (Figure 2C). AcB7-33 was selected as the scaffold for this test as N-terminal acetylation has the potential advantage of providing stability against aminopeptidases. The binding data for these analogs confirm the important role that R13, R17, and I20 (R^B13^, R^B17^, and I^B20^ in the native two-chain H2 relaxin) play in the binding affinity to the RXFP1 ectodomain. The alanine-substituted B7-33 variants highlight the utility of the HEK-7BP competitive binding assay to resolve B7-33 analog binding affinity changes.

### 2.2. Identification of a Minimal B7-33-Based Analogue

With the capacity to differentiate B7-33 analogs using the HEK-7BP competitive binding assay efforts were then made to further reduce the length of B7-33 to a minimal active sequence and assess the binding affinity in this assay (Figure 3). Truncation of the C-terminus using two residues was tolerated in the B7-31 analogs (Figure 3); however, further reduction dramatically affected the 7BP binding (B7-29 Figure 3). When truncating from the N terminus, the scaffold is less amenable to truncation, with the removal of as little as two residues reducing binding capacity (B9-31, Figure 3). Importantly, the cAMP activity of these analogs in the HEK-RXFP1 cells closely matched the 7BP binding affinity data (Figure 3B). These data suggest that B7-33 remains the minimal active sequence. However, B9-31 represents a suitable construct on which to attempt activity recovery to yield a shorter active scaffold.

### 2.3. Aminoisobuteric Acid Substitutions

The incorporation of non-proteogenic, alpha-disubstituted amino acids, such as 2-aminoisobuteric acid (Aib), has been shown to induce peptide helicity [27], which would be beneficial in our search for a minimal H2 relaxin scaffold. We chose to attempt pKi improvement of the B9-31 H2 relaxin analog as this peptide has significant room for improvement within the bounds of the HEK-7BP assay [28] (Figure 4). Here, we explored the substitution of Aib in B9-31, whereby the positions were selected based on SAR data that suggest these positions are not important for RXFP1/7BP binding. Interestingly, any incorporation of Aib reduced the pKi (Figure 4C). We suspected that Aib would induce a helical propensity to B9-31, which we had hypothesized would improve pKi; however, when assayed for helicity in aqueous and TFE buffers, we observed a reduction in helicity through Aib incorporation (Figure 4C–E). This preliminary Aib scanning data suggest that Aib’s propensity to induce helicity is sequence- or position-specific and may not always serve to induce helicity in a target peptide. 

### 2.4. Hydrocarbon Stapling Effect

In our previous studies on the related peptide human relaxin-3 (H3 relaxin), we have seen dramatic improvements in the potency of single-B-chain analogs following the incorporation of a hydrocarbon (HC) staple [29]. Here, we explored HC staples on B7-33 between positions 10 and 14, 14 and 18, and 22 and 26, through substitution with (S)-2-(4-pentenyl) alanine and on resin ring-closing metathesis (RCM) achieved with a Grubbs catalyst (Figure 5). Unlike our previous work on the H3 relaxin B chain [29], the rigid HC staple dramatically reduced the binding affinity of B7-33 analogs. Our data confirm the previous findings from the Rovero group, where they showed triazole conformationally constrained B-chain analogs of H2 relaxin exhibit significantly reduced RXFP1 activity compared with non-stapled B7-33 [30]. These results suggest that B7-33 requires a certain degree of flexibility to adopt a high-affinity binding state for RXFP1/7BP.

## 3. Materials and Methods

### 3.1. Peptide Synthesis and Purification

Standard Fmoc peptide synthesis protocols were used with 4-fold molar excess of the Fmoc-protected amino acids in the presence of 4-fold HCTU (GLBiochem, Shanghai, China) and 8-fold DIPEA on a Biotage Initiator + Alstra microwave synthesizer (Biotage, Uppsala, Sweeden). The protected peptides containing S5 ((S)-2-(4-pentenyl)alanine) (GLBiochem, Shanghai, China) were subjected to on-resin RCM with Grubbs catalyst (Merck, Darmstadt, Germany) according to the method previously reported [31]. After synthesis, the peptides were cleaved from the solid support using a cleavage cocktail of 95% TFA: 2.5% TIPS: 2.5% H_2_O for 2 h at room temperature. After cleavage, the resin was removed through a cotton filter, and the eluate was transferred to a fresh vessel before blowing off the TFA under a continuous flow of N_2_. The scavengers were removed with 2 successive washes with ice-cold diethyl ether (Merck, Darmstadt, Germany) where the ether precipitated the peptide, which was collected by centrifugation at 4000× *g* for 5 min before decantation of the diethyl ether. The peptides were then purified using reverse-phase high-performance liquid chromatography (RP-HPLC), using a Shimadzu (Kyoto, Japan) LC-20AP with an SPD-20ABLK detector using a column oven at 40 °C with solvents of water and acetonitrile each containing 0.1% TFA and using a gradient of 10–60% acetonitrile in 60 min (Merck, Darmstadt, Germany), with LC solution software (Shimadzu, Kyoto, Japan). The final products were characterized using RP-HPLC over 20 min or 30 min from 0–100% acetonitrile as well as with matrix-assisted laser desorption/ionization time of flight mass spectrometry (MALDI-TOF MS) on a Shimadzu (Kyoto, Japan) MALDI-8020 MALDI-TOF mass spectrometer (Appendix A). After achieving >95% purity as determined with RP-HPLC, the peptides were assessed as per the methods in Appendix A. 

### 3.2. Competitive Binding Assays in HEK-RXFP1 and HEK-7BP Cells

Human embryonic kidney (HEK-293T) cells stably transfected with RXFP1 were cultured in RPMI 1640 medium supplemented with 10% fetal calf serum, 100 μg mL^−1^ penicillin, 100 μg mL^−1^ streptomycin, and 2 mM of l-glutamine and plated into 96-well plates pre-coated with poly-l-lysine for whole-cell binding assays (Merck, Darmstadt, Germany). Competition binding experiments using Eu^3+^-labeled H2 relaxin in the absence or presence of increasing concentrations of unlabeled H2 relaxin B-chain derivatives were conducted as previously described [13]. All data are presented as the mean ± S.E.M. of the % specific binding of triplicate wells, repeated in at least three separate experiments, and curves were fitted using one-site binding curves in GraphPad Prism 9 (GraphPad Inc., San Diego, CA, USA). Statistical differences in pKi values were analyzed using a one-way analysis of variance with uncorrected Fisher’s least significant difference (LSD) post hoc analysis in Graphpad Prism 9.

### 3.3. cAMP Activity Assays in HEK-RXFP1 Cells

The ability of peptide analogs to activate cAMP in HEK-RXFP1 cells stably transfected with a pCRE β-gal reporter gene construct was tested as described in detail previously [13]. Cells were stimulated with increasing concentrations of peptide analogs in parallel to H2 relaxin (positive) or media (negative) controls, then incubated at 37 °C for 6 h, after which the media were aspirated, and the cells were frozen at −80 °C overnight. The following day, cAMP-driven β-gal expression was determined in cell lysates as described [32]. Experiments were performed in triplicate at least 3 times, and data were fitted to a four-parameter sigmoidal dose–response curve using GraphPad Prism 9 to determine ligand potency (pEC50).

### 3.4. Circular Dichroism Assay

The peptide secondary structure was estimated using CD spectroscopy [33]. Peptides were assessed at 50 µM in aqueous solutions (100 mM NaF, 10 mM KH_2_PO_4_, and pH 7.5) and a 50% trifluroethanolamine (TFE) solution with the same salts as the aqueous solution (Merck, Darmstadt, Germany). CD spectra were collected from 3 scans at 1 nM intervals between 190 nm and 260 nm using a Jasco J-815 circular dichroism spectropolarimeter (JASCO, Toyko, Japan). Blanks were subtracted and samples were normalized at 260 nm before the mean residual ellipticity (θ) was calculated using the formula: θ = mdeg/(c, I, Nres), where mdeg is the CD output as described above in millidegrees, c is the molar concentration of the peptide, l is the light path length in mm through the sample, and Nres is the number of residues. The percentage of helical residues on average was calculated using the formula Ha = (q222 nm-qC)/(qN222 nm-qC),where q222 nm is the lowest value between 218 and 222 nm, qC = 2220 − 53 T, and qN222 nm = (−44,000 + 250 T)(1 − k/Nres) with T in _C and k = 3.0 [34].

## 4. Conclusions

In our investigations, we have found that our previous single-chain H2 relaxin analog, B7-33, cannot be further reduced without negatively affecting the RXFP1/7BP binding affinity, as determined in our HEK-7BP binding assay. B9-31, or another truncated analog, better serves our interest in developing potential therapeutic peptides as the reduced size is intrinsically better suited to stabilization efforts. The finding that Aib incorporation reduced helical propensity was unexpected, suggesting that sites of incorporation are important to determine the effect on the helicity of Aib residues. Further, the data presented here and elsewhere [30], that rigid stapling strategies are not suitable for single-chain H2 relaxin analogs, suggest that more flexible stapling strategies should be explored in the future.

## Figures and Tables

**Figure 1 ijms-24-12670-f001:**
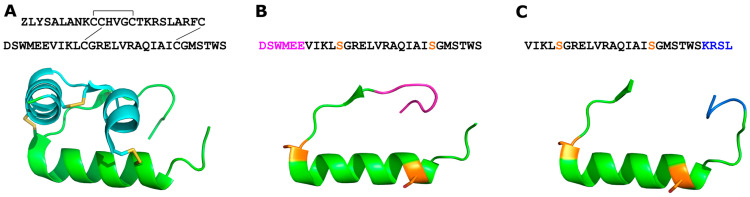
Reduction in H2 relaxin to B7-33. (**A**) NMR structure of H2 relaxin (PDB: 2MV1), with the A-chain shown in blue and B-chain shown in green; the A-chain contains one intra-chain disulfide bond and is attached to the B-chain via two inter-chain disulfide bonds. (**B**) Intermediate B-chain peptide, containing C11S and C23S modifications; the N-terminal sequence that is removed is colored pink to generate B7-29. (**C**) B7-33, with the B1-33 H2 relaxin C-terminal residues (KRSL), is shown in blue for clarity.

**Figure 2 ijms-24-12670-f002:**
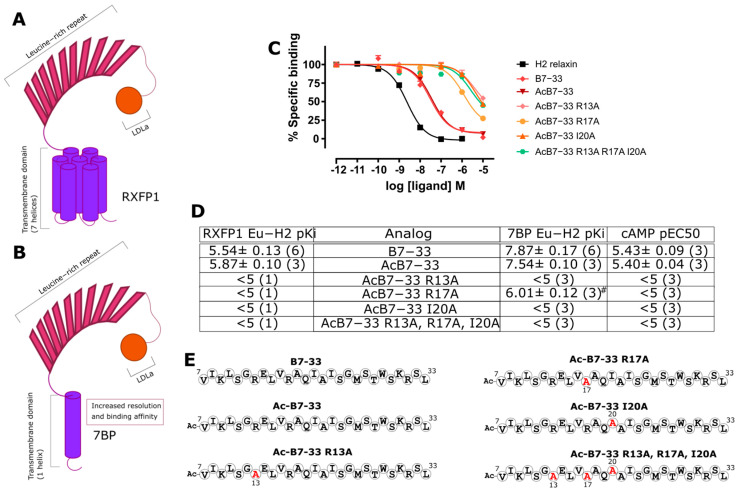
RXFP1, 7BP, and binding affinities of alanine analogs of B7-33. Cartoons depicting (**A**) RXFP1 compared to (**B**) 7BP, which consists of the RXFP1 ectodomain fused to a single transmembrane domain. (**C**) HEK-7BP binding curves for B7-33 key residue substitutions. (**D**) Acetylated B7-33 analogs with alanine substitutions (in red) at key binding residues were compared across RXFP1 and 7BP, which identified 7BP as able to resolve the difference of binding due to each residue substitution. (**D**) Representation of B7-33 with positions of substitutions identified. (**E**) Peptide sequences of the analogs tested. ^#^ *p* < 0.001 vs. B7-33.

**Figure 3 ijms-24-12670-f003:**
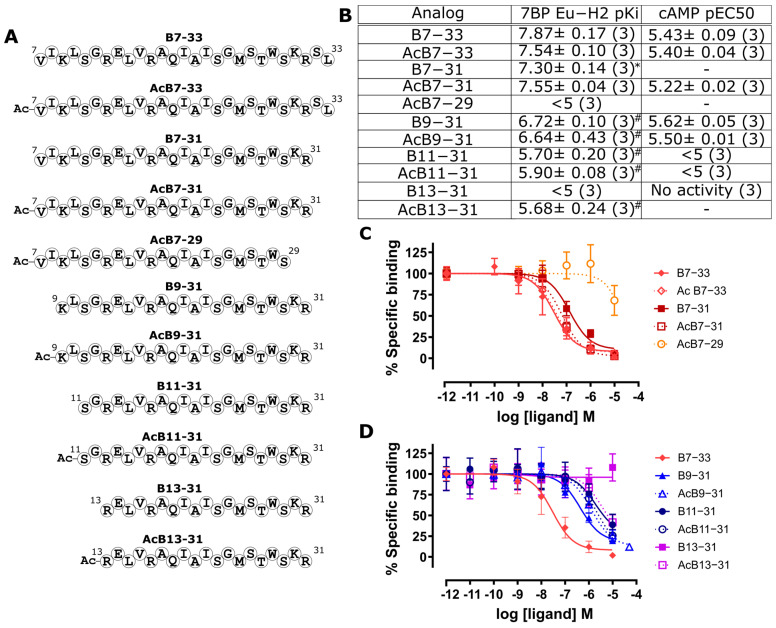
Attempts to reduce the single-chain H2 relaxin scaffold sequence length. (**A**) Representation of H2 relaxin single-chain analog sequences. (**B**) pKi data for HEK-7BP binding (from C/D) and cAMP potency data; pEC50 for RXFP1 (from Appendix A) for analogs. (**C**) HEK-7BP binding curves for C-terminal truncations. (**D**) HEK-7BP binding curves for N-terminal truncations. * *p* < 0.05 vs. B7-33 and ^#^ *p* < 0.001 vs. B7-33.

**Figure 4 ijms-24-12670-f004:**
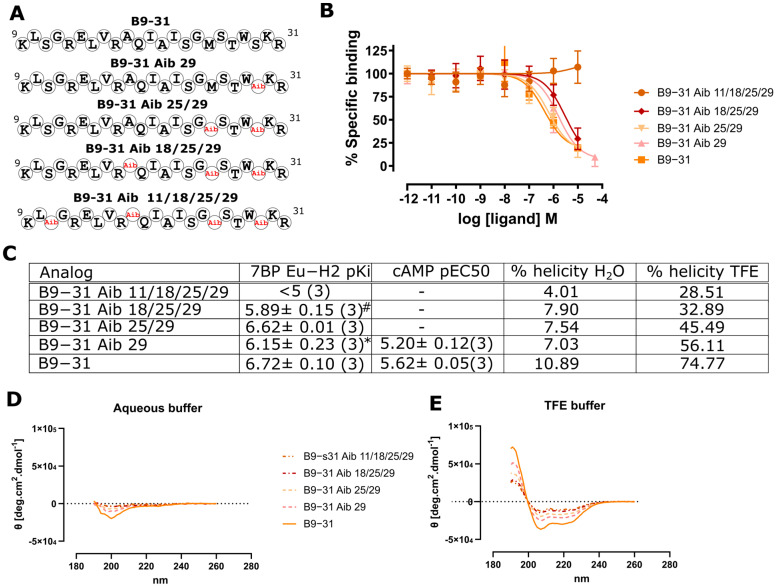
Aib incorporation into B9-31. (**A**) position of Aib incorporation into B9-31 (red). (**B**) 7BP competitive binding assays on analogs. (**C**) The effect of Aib incorporation on Eu-H2 relaxin binding affinity pKi (from (**B**)) and cAMP potency pEC50 (from Appendix A). Analog helicity in (**D**) aqueous and (**E**) TFE-containing phosphate buffers. * *p* < 0.01 vs. B9-31 and ^#^ *p* < 0.001 vs. B9-31.

**Figure 5 ijms-24-12670-f005:**
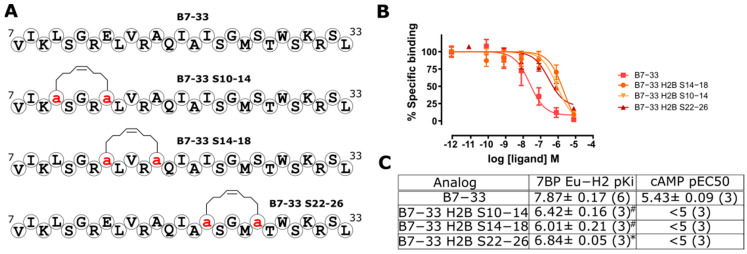
HC-stapled B7-33 analogs. (**A**) Sequences of HC-stapled B7-33 analogs with positions of (S)-2-(4-pentenyl) alanine are shown as a red ‘a’. (**B**) 7BP competitive binding curves of analogs. (**C**) Eu-H2 relaxin binding affinity, pKi (from (**B**)), and cAMP potency pEC50 (from Appendix A). * *p* < 0.01 vs. B9-31 and ^#^ *p* < 0.001 vs. B7-33.

## Data Availability

All the data are the result of *n* = 3–4 independent experiments and are expressed as mean ± SEM. We used GraphPad Prism 9 to make graphs and statistically analyze data.

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
