# Peer review of "Further Developments towards a Minimal Potent Derivative of Human Relaxin-2"

_ijms, 2023, doi:10.3390/ijms241612670_

Round 1
Reviewer 1 Report
The manuscript is very interesting and well-written.
I have no issues regarding manuscript writing.
However, the fonts and sizes of the text in all figures are not consistent.
Some are big, some are too small, and some have different fonts and formats.
Please fix all the figures before the publication process.
Overall, I believe it is suitable for publication in IJMS as a communication and can be published after fixing all the text fonts in all figures.
Author Response
We would like to thank Reviewer #1 for their time taken to review our manuscript and for their helpful comments.
We have amended the figures as requested.
Reviewer 2 Report
The Authors decided to verify the minimal length of Human Relaxin 2 analogue which maintains the binding affinity to RXFP1. The idea of shortening the peptide chain of H2 relaxin is excellent, because chemical synthesis of peptides containing 3 disulfide bridge is an enormous challenge.
Unfortunately, the results obtained in this study prove that the B7-33 peptide, which has already been published, cannot have shorter peptide chain while maintaining RXFP1/7BP binding affinity. However, the Authors describe the relevance of the study and highlight the fact, that incorporation of 2-aminoisobuteric acid (Aib) does not always induce peptide helicity.
I suggest only minor corrections considering graphical part of the manuscript:
- the tables are not readable. Please change the design of the table, interline, bold the headings, or etc.
- the font size of axes units and titles in Fig. 3C,D and 4B are definitely too big.
Author Response
We would like to thank Reviewer #2 for their time taken to review our manuscript and for their helpful comments.
We have amended the tables for legibility, bold for the headings as requested. Figures 3C,D and 4B have been made smaller as requested.
Reviewer 3 Report
In the present manuscript, Thomas et al. detail their efforts to enhance the effectiveness of the previously developed human relaxin 2 peptide. To achieve this goal, the researchers explored strategies to reduce the peptide's binder size. Remarkably, during their investigation, the authors discovered that the B chain of the original peptide exhibited comparable binding affinity. Subsequently, the team made endeavors to further enhance the peptide's pharmacokinetic properties (PKi) and binding affinity. Two specific approaches have pursued this purpose: the substitution of 2-amino isobutyric acid (Aib) and hydrocarbon stapling through ring-closing metathesis (RCM). Regrettably, these attempts did not yield the desired improvements, as the PKi and binding affinity remained unchanged or lowered.
The findings presented in this manuscript highlight the challenges faced in the optimization of the human relaxin 2 peptide and underscore the importance of carefully considering alternative strategies for future investigations. The outcomes of these endeavors contribute valuable insights and may pave the way for further advancements in the development of potent therapeutic peptides. The manuscript should be suitable for publication after addressing the following concerns:
1. Did the authors examine the binding affinity with chain A of the parent peptide?
2. Can authors perform alanine scanning to check the contribution of each reside within the sequence?
3. Why didn't the authors check the RRI replacements for Aib-substituted peptides in Figure 4?
4. Why didn't the authors perform the stapling for B9-31 instead of B7-33 peptides in Figure 5?
5. Can the authors provide the HPLC and Mass data for all synthesized peptides?
Author Response
We would like to thank Reviewer #3 for their time taken to review the manuscript and their comments. Our responses to their comments are here:
- The A chain has previously been shown to not contribute to binding as shown in our earlier, referenced manuscript – ref 32
- H2 Relaxin the parent peptide for B7-33 has undergone extensive SAR studies, a review article on this has been referenced – ref 25, We have confirmed in reference 32 and here that R13, R17 and I20 are the key residues for RXFP1/7BP binding
- The goal of the Aib substituted peptides in figure 3 was to improve the binding not to reduce the binding. As we saw loss of binding for these analogs it was not necessary to test RRI replacements in the Aib substituted analogues.
- Our attempts in figure 3 lead us to believe that the affinity of b9-31 could not be improved, thus we attempted the stapling in figure 4 on the parental compound b7-33.
- Some analogues are previously published B7-33, Ac-B7-33 and the R13A, R17A, I20A analogues and this data is available in ref 32. Other analogues have been added into the supplemental data section for both HPLC traces and MS data.
Round 2
Reviewer 3 Report
The authors have addressed all the queries and added the required analytical data as requested by the reviewer. Therefore this manuscript is now suitable for publication in its current form.